

# Multimillennial synchronization of low and polar latitude ice cores by matching a time constrained Alpine record with an accurate Arctic chronology

Paolo Gabrielli[1,2], Theo M. Jenk[3,4], Michele Bertó[5], Giuliano Dreossi[6], Daniela Festi[7], Werner Kofler[7],

Mai Winstrup[8], Klaus Oeggl[7], Margit Schwikowski[3,4,9], Barbara Stenni[5] and Carlo Barbante[5,6]

[1]Byrd Polar and Climate Research Center, The Ohio State University, Columbus, 43210, USA

[2]School of Earth Sciences, The Ohio State University, 275 Mendenhall Laboratory, Columbus, 43210, USA

[3]Laboratory of Environmental Chemistry, Paul Scherrer Institut, 5232, Villigen PSI, Switzerland

[4]Oeschger Centre for Climate Change Research, University of Bern, CH-3012 Bern, Switzerland

[5]Department of Environmental Sciences, Informatics and Statistics, Ca' Foscari University of Venice, Venice-Mestre, 30170, Italy

[6]Institute of Polar Sciences-CNR, Venice-Mestre, 30170, Italy

[7]Institute for Botany, University of Innsbruck, Innsbruck, 6020, Austria

[8]DTU Space, Technical University of Denmark, Kongens Lyngby, Denmark

[9]Department of Chemistry and Biochemistry, University of Bern, CH-3012 Bern, Switzerland

*Correspondence to:* Paolo Gabrielli (gabrielli.1@osu.edu)

**Abstract.** We present a novel application of empirical methodologies that significantly reduce the chronological uncertainty of a low latitude-high altitude Alpine ice core record obtained in 2011 from the glacier Alto dell'Ortles (3859 m, Eastern Alps, Italy). A preliminary absolute timescale based on a peak in $^3$H activity, and $^{210}$Pb and $^{14}$C analyses on carbonaceous particles and organic remains provided evidence of one of the oldest Alpine ice core records spanning the last ~7000 years, back to the last Northern Hemisphere Climatic Optimum. Here we combine three empirical methods that provide an additional number of time markers that corroborate the multimillennial nature of the Alto dell'Ortles ice cores while significantly decreasing the uncertainty of the chronology. First, $^{14}$C analysis of an additional organic fragment (a charred spruce needle) discovered next to the basal ice provides an age (232 ± 126 BCE) which agrees with previous $^{14}$C dates in the oldest part of the record. Second, a new millennial-scale high resolution atmospheric Pb depositional record was used to synchronize the Alto dell'Ortles cores



with an accurately-dated (±5 years) Pb record from an array of Arctic ice cores during the ~200 BCE to ~1900 CE time period. This match resulted in a shift of the initial Alto dell'Ortles timescale ~200 years earlier, but still within the initial time uncertainty. Third, novel seasonally resolved pollen records from the upper firn/ice portion of the Alto dell'Ortles cores were combined with $\delta^{18}O$ and dust annual variations to refine the dating for the 20$^{th}$ century by means of an automatic algorithm (Straticounter; between 2011 and 1927 CE) and visual counting (1926-1900 CE). The time markers obtained by these three methods were combined in a continuous timescale by running a Montecarlo based fit (COPRA model). This Alto dell'Ortles revised chronology shows a significantly reduced uncertainty, between ±1 and ±4 years after 1927 CE, and between a maximum of ±100 years to a minimum of ±5 years between 1927 CE and 200 BCE by conservative estimates. An investigation of the revised chronology by means of a simple 1-D flow model suggests that non-steady-state conditions (e.g., changes in past snow accumulation rate) need to be considered to provide a full physical explanation of the age-depth relationship obtained. The new revised chronology will allow the constraint of the Holocene climatic and environmental histories emerging from this high-altitude glacial archive of Central Europe. The novel methodologies may also be adopted to build or improve the chronologies of other ice cores extracted from-low latitude/high-altitude glaciers that typically suffer from larger dating uncertainties compared with well dated polar records.

## 1 Introduction

Ice cores extracted from polar regions and high-altitude/low-latitudes glaciers are archives of past climatic-environmental histories as they record physical, chemical and biological characteristics of the past atmosphere. In order to interpret this information, it is of fundamental importance to link the englacial depth of the various ice sections to the timing of the original snow deposition. This provides a function that is commonly known as chronology.

Counting annual layers, establishing time markers (e.g. volcanic horizons, $^{10}$Be, $^{3}$H peaks etc.), synchronizing with other dated paleo-records, and developing ice flow models are widely used methods to precisely and accurately date ice cores from polar regions (Parrenin et al., 2007; Svensson et al., 2020), where the negligible/slow horizontal flow at the drilling sites and the largely below freezing englacial temperatures allow a full physical-chemical preservation of the ice stratigraphy and the time markers embedded within the accumulated snow layers. In contrast, dating ice layers from high-altitude/low-latitude glaciers is more challenging because of: i) their smaller ice thickness, implying that the portion including annual layers is usually limited between the upper 50 to 100 meters (typically covering only a few centuries) (Schwikowski et al., 2014); ii) a larger horizontal flow at the drilling sites that can quickly alter/disrupt the original ice stratigraphy (Thompson et al., 2000), and; iii) increasingly widespread post depositional processes that are linked to modern warmer air temperature and summer meltwater percolation which can overprint (or prevent the full conservation of) the time markers embedded in the glaciers (Gabrielli et al., 2010).

In 2011 we extracted four 60-75 meter long ice cores from the Alto dell'Ortles glacier (3859 m) near the summit of Mt. Ortles (3905 m) in the Eastern Alps, Italy. We demonstrated that the upper firn portion of the Alto dell'Ortles glacier was temperate, with intense summer meltwater percolation that possibly affected this drilling site since the 1980s. However, the



underlaying ice was still cold (-2.8 °C near the bedrock) (Gabrielli et al., 2012) and preserved stratigraphy that was thousands of years old (Gabrielli et al., 2016). Our discovery of millennial-age ice in the Eastern Alps was recently confirmed by another ice core record extracted from the nearby drilling site of Weißseespitze (3500 m, Austria) (Bohleber et al., 2020).

In order to develop an initial chronology, in 2016 we identified well-defined $^3$H and beta activity peaks at 41 m depth attributed to 1963 atmospheric thermonuclear testing and used the activity of $^{210}$Pb to date the Alto dell'Ortles cores to 59 m depth (~1930 CE). In addition, determination of a $^{14}$C age in a larch needle embedded near the basal ice, along with $^{14}$C dating of several ice samples (using the water insoluble organic fraction of carbonaceous aerosols entrapped; WIOC-14) allowed us to obtain absolute time markers from the deepest ice, the age of which was calculated date back to ~7000 years. All the age constraints obtained were used to build an initial chronology (TC2016 from now on) by means of a Montecarlo fit from the COPRA model (Breitenbach et al., 2012), which also provides time uncertainty as a function of depth (from a few years in the upper modern firn layers to ± 500 years in the deepest portion of this ice core (Gabrielli et al., 2016)).

A large uncertainty in TC2016 occurred not only in the basal layers of the ice core but also in the intermediate depths (60-72 m), dated between 1900 and 200 CE (e.g., ~ ± 50 years at 1850 CE, ~ ± 100 at 1650 CE and ~ ± 400 at 500 CE). The relative time uncertainty (50-70%) of the intermediate portion of the record was higher than both the modern (1900 -2011 CE, 5-25% uncertainty) and the oldest (200 CE – 5000 BCE; 10-30% uncertainty) sections. This would limit the ability to reconstruct precise climatic and environmental histories of the Alps and Central Europe from these cores. The large uncertainty in the intermediate portion of the cores was essentially due to: i) the lack of time markers such as volcanic horizons; and ii) the use of only two WIOC-14 $^{14}$C ages (1355 ± 205 CE and 429 ± 286 CE) above 72 m depth (200 CE). These measurements were also characterized by relatively large uncertainty due to the intrinsic limits of the $^{14}$C technique to date organics that were centuries to millennia old (Uglietti et al., 2016).

Here we report additional stratigraphic/time markers, the use of which confirms TC2016 within the large time uncertainty initially presented, and shift back in time the intermediate part of the ice core record by ~200 years, significantly reducing the time uncertainty. This revision was performed by first refining the depth alignment of the three Alto dell'Ortles cores, for which a greatly increased number of tie points obtained from the stable isotopes records was utilized, and then by running the COPRA Montecarlo based model to fit a larger selection of original and new time markers. In particular, the new age constraints were obtained from: 1) $^{14}$C dating of an additional organic fragment (a charred spruce needle; 232 ± 126 BCE) discovered near the bottom ice (72.82 m) in core #1; 2) the synchronization of the high resolution crustal excess Pb flux record in core #3 (which was substantiated by a second, less resolved Pb record from core #1) to a corresponding well-dated (±5 years) Arctic record from ~1900 CE to ~200 BCE (Mcconnell et al., 2018); and 3) the automated counting of annual layers with Straticounter (Winstrup et al., 2012) based on pollen and $\delta^{18}$O and dust records from core #1 from 2011 to 1927 CE (51 m depth in core #1) and visual counting from 1926 to 1900 CE (57 m depth in core #1). Finally we applied the glaciological ice flow model developed by Dansgaard-Johnsen (Dansgaard and Johnsen, 1969) at the Alto dell'Ortles drilling site to investigate the physical nature of the revised empirical chronology (age-depth relationship).



## 2 Mt. Ortles ice cores re-alignment

The initial chronology, TC2016 (Gabrielli et al., 2016), was developed by employing a common depth scale for Alto dell'Ortles cores #1, #2, and #3, using core #2 depth as a reference. Depth alignment of core #2 with #1 and with #3 was performed by matching common features in their $\delta^{18}$O records (17 between #2 and #1, 14 between #2 and #3) (see Fig 10 in Gabrielli et al., 2016 ). This allowed the transfer of time markers in the cores to a common depth scale. Because of the lower resolution of the $\delta^{18}$O record in core #3, only two match features, or "tie points", could be established between cores #3 and #2 below 60 m depth. However, we realized later that this resulted in a depth misalignment of a only few tens of cm, which nevertheless resulted in a significant time lag (up to ~300 years) between the 3 cores. This mismatch in depth and time became apparent by comparing two new high resolution Pb concentration records from cores #1 and #3 (see below) with TC2016. The much larger Pb variability below 60 m depth, compared to the $\delta^{18}$O variability, facilitated the detection of the initial mismatch of the isotope records in the ice cores.

For this work, the first step to develop a more accurate chronology was to revise the common depth scale for the three cores, still using core #2 for depth reference. Depth realignments of core #2 with cores #1 ad #3 were performed in two steps using the Analyseries 2.0.8 software (Paillard et al., 1996). First, the $\delta^{18}$O records were more finely matched using additional tie points between cores #1 and #2 (for a total of 122 points) and between cores #3 and #2 (87 points). However, while a more detailed alignment was obtained (Fig. 1, black vertical lines), a lack of tie points persisted within the 60-73 m depth interval between cores #2 and #3.

To better align the $\delta^{18}$O #2 and #3 records in this interval, 31 additional tie points were obtained from two high resolution Pb concentration records independently determined in cores #1 and #3 by discrete Inductively Coupled Plasma Mass Spectrometry (ICP-MS) and continuous flow analysis  ICP-Sector Field MS (ICP-SFMS) at the University of Venice and at The Ohio State University, respectively (Gabrieli, 2008; Gabrieli and Barbante, 2014). Pb was not analysed in core #2 due to insufficient ice volume. Depth alignment between 60 and 73 m was performed in three steps: i) by matching the Pb concentration records in cores #1 and #3 (31 tie points obtained) using Analyseries 2.0.8 software (Paillard et al., 1996) (Fig. 2); ii)  by transferring the depths of the 31 Pb points in core #1 to core #2 using the depth map obtained by matching cores #1 and #2 using $\delta^{18}$O, see above; and iii) by linking the $\delta^{18}$O stable isotope records in cores #2 and #3 over the interval using as a guide the depth of the 31 supplemental Pb tie points (Fig. 1, red vertical lines).





Figure 1: Revised depth alignments using the δ¹⁸O records from the Alto dell'Ortles cores #1, #2 and #3 over the entire lengths (upper panel) and the bottom portions (lower panel) of the cores. Core #2 is used as reference depth (X axis). Black vertical links indicate the δ¹⁸O tie points while red links those ties obtained through the Pb records of cores #1 and #3 (see Fig. 2). Adapted from Fig. 10 in (Gabrielli et al., 2016).



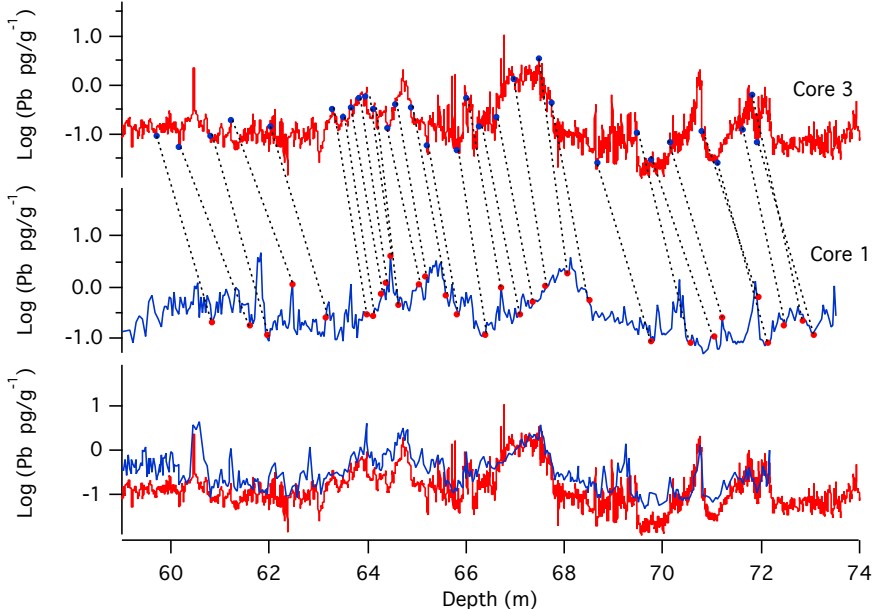

Figure 2: Pb concentrations in the deep portion of cores #3 (red; Continuous Flow Analysis ICP-SFMS, The Ohio State University) and #1 (blue; discrete ICP-MS analysis, University of Venice) in their respective #1 and #3 depth scales (upper two panels) and matched to core #3 depth (lower panel).

## 3 Time markers

After the more precise alignment (synchronization) of the three Alto dell'Ortles cores, we selected old time markers from TC2016 and new markers to build the revised chronology. In the deepest portion we retained the five oldest [14]C ages before 442 CE and added a new [14]C age (232 ± 126 BCE) (see paragraph 3.1; Table 1). Between 171 BCE and 1907 CE, 21 additional time markers were obtained from a match between the crustal excess Pb flux record in the Alto dell'Ortles core #3 and the well-dated (±5 years) Severnaya Zemlya Arctic core (McConnell et al., 2018) (see paragraph 3.2, Supplementary Table 1). The environmental interpretation of the Pb record is not within the scope of this paper and will be discussed in detail in a separate manuscript. Finally, from 1900 to 2011 CE, while we retained the beta and [3]H activity peaks of 1954 and 1963 CE from TC2016, we added three additional time constraints (2006, 1995, 1986 CE; Table 2) and 113 new time markers obtained by instrumental and visual annual layer counting of pollen, dust and $\delta^{18}$O (Paragraph 3.3; Supplementary Table 2).

### 3.1 The new [14]C time marker near the bottom ice (232 BCE)

An intact larch needle at 73.25 m in core #1 provided a [14]C calibrated age of 659 ± 102 BCE (Gabrielli et al., 2016), which was useful for dating the older part of TC2016. More recently we extracted a spruce charred needle from core #1 at 72.82 m (Fig. 3). Spruce are very common conifers in the Alps, including the Mt. Ortles area. Atmospheric vertical convection





most likely transported the two needles from lower elevation to the Alto dell'Ortles glacier. The charring of the needle likely resulted from a local forest fire, the heat from which may have produced or intensified atmospheric convection. This fragment, which contained 57 μg of carbon, was radiocarbon dated at the Paul Scherrer Institute at the Bern AMS facility (LARA Laboratory, University of Bern, Switzerland) (Szidat et al., 2014) and provided a conventional $^{14}$C calibrated age of $232 \pm 126$

5    BCE (Bern AMS sample number BE-12451.1.1). This date is stratigraphically consistent with the $^{14}$C date ($659 \pm 102$ BCE) of the first larch needle found at 73.25 m in core #1 and also with the shallower/younger and deeper/older WIOC-14 dates (see Table 1). Within the range of uncertainty, the $^{14}$C calibrated age ($232 \pm 126$ BCE) of the charred needle found at 72.82 m is consistent with the TC2016 dating ($146 \pm 370$ BCE) at the same depth.

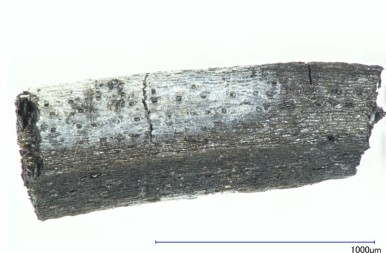

10   Figure 3: The spruce charred needle fragment found in core #1 at 72.82 m of depth and $^{14}$C dated at $232 \pm 126$ BCE. The image was produced by using a Kayence VHX 2000 digital microscope at the University of Innsbruck.

| Core # | Tube # | Measure | Top depth (m) | Bottom depth (m) | WIOC-14 (µg) | F$^{14}$C | $^{14}$C age (yrs BP) | Cal age (yrs cal BP) | µ cal age (yrs cal BP) | µ cal age (yrs cal b2012) | µ cal age (CE-BCE) | σ (yrs) | TC2016 | This study |
|---|---|---|---|---|---|---|---|---|---|---|---|---|---|---|
| 1 | 98b | WIOC-14** | 68.26 | 68.96 | 50.03 | 0.927 ± 0.025 | 609 ± 217 | (330 - 784) | 589 | 651 | 1361 | 204 | ✓ | |
| 3 | 102 | WIOC-14** | 70.87 | 71.57 | 28.41 | 0.823 ± 0.027 | 1565 ± 264 | (1178 - 1781) | 1508 | 1570 | 442 | 288 | ✓ | ✓ |
| 1 | 103b | WIOC-14 | 71.77 | 72.48 | 10.37 | 0.932 ± 0.037 | 569 ± 320 | (155 - 903) | 566 | 628 | 1384 | 289 | | |
| 1 | 104b | Charred spruce needle | 72.78 | 72.82 | 57* | 0.761 ± 0.009 | 2193 ± 98 | (2333-2070) | 2182 | 2244 | -232 | 126 | | ✓ |
| 1 | 105b | Larch needle | 73.25 | 73.25 | 68* | 0.728 ± 0.006 | 2550 ± 65 | (2499 - 2751) | 2609 | 2671 | -659 | 102 | ✓ | ✓ |
| 3 | 106 | WIOC-14 | 73.73 | 74.02 | 10.91 | 0.628 ± 0.031 | 3737 ± 397 | (3578 - 4786) | 4171 | 4233 | -2221 | 524 | ✓ | ✓ |
| 3 | 106 | WIOC-14 | 74.02 | 74.24 | 11.50 | 0.568 ± 0.030 | 4544 ± 424 | (4620 - 5718) | 5176 | 5238 | -3226 | 531 | ✓ | ✓ |
| 3 | 106 | WIOC-14 | 74.24 | 74.47 | 18.47 | 0.481 ± 0.020 | 5879 ± 334 | (6321 - 7156) | 6739 | 6801 | -4789 | 364 | ✓ | ✓ |

*Pure C extracted after combustion
** Combined values from the three sub-samples of tubes 98b and  102

Table 1: $^{14}$C analysis of organic material from the Ortles cores (adapted from Table 2 in Gabrielli et al., 2016). The radiocarbon dates (µ cal age; the most probable age) were calibrated using IntCal20. Tube # 104b and 105b in core #1 refer to the locations of the two conifer needles found in the bottom ice. All other $^{14}$C ages were obtained by measuring the water insoluble organic fraction of carbonaceous aerosols entrapped (WIOC-14) in various ice sections of the Alto dell'Ortles cores #1 and #3. For the

20   selection of these data for the revised time scale see text and our previous publication (Gabrielli et al., 2016).



3.2 Millennial synchronization with the Arctic Pb record (171 BCE - 1907 CE)

Well-dated Pb records from six Arctic cores (McConnell et al., 2018) show similarities with the high resolution Pb record from the Alto dell'Ortles ice core since ~200 BCE. Among the Arctic Pb records, the one from Severnaya Zemlya (SZ; Russian Arctic) was selected for synchronization with the Alto dell'Ortles cores (Fig 4) because: i) it contains one of the
longest continuous chronology; ii) it provides the largest Pb amplitude signal; iii) it is suggested by atmospheric modelling to be the most influenced by Pb emissions from Ag mining and metallurgical activities in central Europe close to Mt. Ortles during the past millennia (McConnell et al., 2018), making it more likely that the Alto dell'Ortles and SZ drilling sites share similar variability in atmospheric Pb depositions. The uncertainty of the SZ chronology was estimated to be less than 5 years between 500 and 1999 CE, with less than 2 years uncertainty at the known volcanic time horizons with the Greenland reference
record (McConnell et al., 2018).

The continuous flow analysis record from Alto dell'Ortles core #3 (Fig. 2) was selected for synchronization with SZ because of its higher time resolution. Synchronization was performed using the Analyseries 2.0.8 software (Paillard et al., 1996) to match the non-crustal Pb flux between SZ and Alto dell'Ortles (see below). This flux in SZ was selected as it is independent from the local snow accumulation and the geological background while it likely retains the variability of the
atmospheric anthropogenic Pb, the signal of which was suggested to have a continental/European significance in this sector of the Northern Hemisphere before the 20[th] century (McConnell et al., 2018). During the 20[th] century different local sources from Russia and Central Europe may have played a role in decoupling the trends of the SZ and Alto dell'Ortles Pb records. Therefore, synchronization of the non-crustal Pb fluxes was not performed during the 20[th] century portion of the record and annual layer counting in the core #1 was preferred instead (see below). In core #3 the non-crustal Pb flux was obtained by
using the Pb/Rb ratio from the average terrestrial dust composition (Wedepohl, 1995) and an assumed constant snow accumulation rate of 1 m y[-1]. Note that this flux to Alto dell'Ortles is just a linear transformation of the Pb concentration that remains independent of any time scale.

Core #3 was synchronized to SZ by using 22 non-crustal Pb flux ties selected between 171 BCE and 1907 CE (Fig. 4, and Supplementary Table 1). This comprises a time period of more than 2000 years, covering 57 to 72 m. When core #2
was matched to cores #1 and #3, the ages/depths of the two WIOC-14 $^{14}$C dated samples from cores #1 and #3 used in TC2016 (1361 ± 204 CE at 68.43 m; 442 ± 288 CE at 71.84 m, both depths in core #2) are revised to 1096 and 253 CE at these depths, respectively. When compared to TC2016, these revised dates are 265 and 189 years older, which is close to or within the uncertainty intervals (204 and 288 years) of the two calibrated WIOC-14 $^{14}$C dated samples, respectively (Table 1). These two new synchronized ages are also within the combined uncertainty (± 250 and ± 350 years, respectively) provided by the COPRA
model used to develop TC2016.

To assign uncertainties to the 22 selected tied ages obtained by matching SZ with Mt. Ortles core #3, a direct transfer of the uncertainty intervals from the SZ Pb record (±5 years) (McConnell et al., 2018) was considered to underestimate the residual age discrepancy (204 and 288 years, see above) with the two WIOC-$^{14}$C dated samples and some possible accidentally inaccurate matches of the Pb records. As a consequence, an uncertainty of 10% of the matched age (expressed in years before

2012) ranging from 10 up to 200 years was assigned over this time interval. Ten years is a lower limit in the same order of the time uncertainty of the deepest sections dated by using annual layers down to 57 m depth in core #1, see below; 200 years is an upper limit in the same order of the mentioned discrepancy with the two WIOC-$^{14}$C dated deeper samples.



5    Figure 4: Synchronization of the non-crustal Pb fluxes in the cores Alto dell'Ortles #3 (red) with reference time series from Severnaya Zemlya (SZ in blue) (McConnell et al., 2018), with 22 tie points shown. The upper panel zooms on the more recent 1500-1900 CE time period.



In general, the match between Alto dell'Ortles and SZ is excellent, except during 1650-1900 CE when the non-crustal Pb flux consistently increases in SZ while that in the Alto dell'Ortles record remains relatively constant (Fig. 4). One possibility is that the constantly low non-crustal Pb flux was caused by increased summer snow erosion on Mt. Ortles during the Little Ice Age (see section 5.4). When comparing SZ and Alto dell'Ortles with a corresponding Pb dated record from Colle Gnifetti

(CG03B; Western Alps) (see info about CG03B in Supplementary Text 1 and Fig. S1), the agreement between the three records is remarkable back to ~1350 CE; however, the CG03B record diverges afterwards (Fig. S2). Overall, this supports the use of the SZ record for synchronisation with Alto dell'Ortles before 1350 CE. More in detail, between ~1350 and ~1800 CE, the agreement is excellent for CG03B and Alto dell'Ortles, whereas it is acceptable only between ~1350 and ~1650 CE for CG03B and SZ. In fact we observe disparities between CG03B and SZ between ~1650 and ~1850 CE, and between ~1800 and ~1900

CE in the CG03B and Alto dell'Ortles comparison (Fig. S3). A good agreement between CG, SZ and Alto dell'Ortles is observed over the 20th century (Fig. S4).

3.3 Annual layer counting (1900-2011 CE)

A clear seasonal pollen signal was reported from the shallow firn temperate layers (10 m depth) of the Alto dell'Ortles

glacier (Festi et al., 2015; Festi et al., 2017). New pollen records of relative and concentration values were obtained from core #1. Despite recent intense summer meltwater percolation, the seasonality of the pollen record appears to be conserved in the firn and also in the lower ice portion. This is demonstrated by pollen, $\delta^{18}O$ and dust within the well-dated 1954-1963 CE interval which is calibrated by beta and $^3H$ activity peaks. These three parameters consistently show high values during the warm seasons (late spring to late summer) and low values during the cold seasons (Fig. 5).

We applied the StratiCounter algorithm (Winstrup et al., 2012) to a combination of pollen concentrations, $\delta^{18}O$ and dust, and three other derived pollen records (see below), to produce an annual-layer-counted timescale from the 2011 glacier surface to 53 m (41 m w.e.) depth. Note that StratiCounter also produces an uncertainty interval for any identified annual-layer. At depth where the StratiCounter analysis became more uncertain, we attempted visual counting down to 57 m (45 m w.e.), where the combined annual signals seemed distinguishable (Fig. S6). In both cases, the age-markers correspond to

winter/spring of each given year as pollen concentration in the atmosphere is high during spring/summer, while little or no pollen is present in winter. This situation is reflected in the corresponding seasonal ice core layers.

The additional three pollen records obtained using StratiCounter were: i) Day Of the Year, which considers 32 pollen types (DOY32) or; ii) 46 types (DOY46) and; iii) a day-match of the pollen compositional record from a local modern pollen monitoring station (Festi et al., 2017). The latter three are weighted as a single line of information by StratiCounter as they are

not independent. Briefly, each of these three records are derived from the different classified pollen types and provide an estimate of the pollen ensemble depositional day-of-year (DOY; values between 1 and 365) based on the observed pollen type spectra composition in the ice core. DOY32 and DOY46 records are based on 32 and 46 pollen types with known blooming time. The day match record has been obtained using 10 years of data collected at the nearby Bolzano airborne pollen monitoring station (~70 km from the drilling site; https://ean.polleninfo.eu/Ean/).



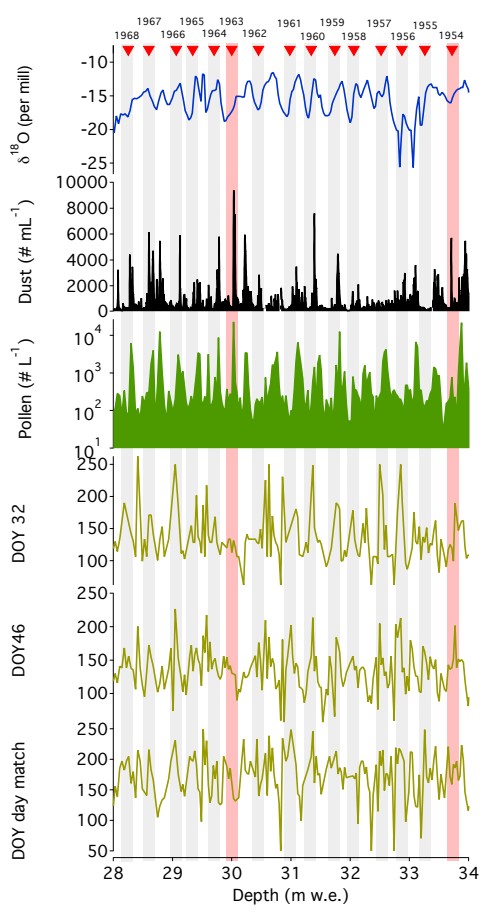

Figure 5: Annual layers between 28 and 34 m water equivalent (w.e) in core #1 shown in $\delta^{18}O$, dust and pollen concentrations, DOY 32 and 46 and DOY variations (see text). Red vertical bars indicate 1963 and 1954 time markers from the $^3H$ and beta activity peaks (Gabrielli et al., 2016). See also supplementary figures S5-S6 that display annual layers in previous periods during the 20th century.

| Core # | Depth #1 (m we) | StratiCounter original timescale (CE) | Adjusted age (CE) | Time marker age (CE) | Time marker |
|---|---|---|---|---|---|
| 1 | 5.98 | 2005.9 | 2006.1 | Spring 2006 | spruce extreme |
| 1 | 13.84 | 1994.9 | 1995.1 | Spring 1995 | spruce extreme |
| 1 | 18.27 | 1986.6 | 1986.8 | 1986 | Beta activity (Chernobyl) |
| 2 | 29.9 | 1960.3 | 1962.5 | 1963 | Beta activity, Tritium |
| 2 | 33.89* | 1952.7 | 1954.9 | 1955 | Beta activity |

*Tranferred from core #2

Table 2: Time markers in the upper portion of the Alto dell'Ortles cores, which were also used to adjust the StratiCounter annual layer counting (e.g. Fig. 5).



The annual layers were compared and adjusted to independent fixed time markers that were identified in core #1 (Table 2). These include three known horizons with high levels of radioactivity: the beta activity peak from the Chernobyl accident in 1986; beta and $^3$H peaks dated at 1963 from atmospheric nuclear tests; and the start of deposition of fallout onto glaciers in 1954 from atmospheric nuclear tests (Gabrielli et al., 2016; Gabrieli et al., 2011). More recently, two additional

time markers were assigned by comparing spruce (*Picea abies*) pollen concentrations with observations from the nearby pollen measuring stations at Innsbruck and Obergurgl (Bortenschlager and Bortenschlager, 2003). The records from these stations are the longest (44 and 41 years, respectively) and contain evidence of three exceptional blooming years in 2006, 1995 and 1992. While the strongest episode in 1992 does not correspond to large spruce concentrations in the Alto dell'Ortles core, the 1995 and 2006 events are characterized by high and broad pollen peaks in the firn portion of the core. The ages of all the

marker horizons fall within the derived uncertainty of the StratiCounter timescale. By using these 5 time marker horizons as guides for adjustments, we removed 2 years between 18.2 and 29.5 m w.e.

We compared the annual time scale obtained by StratiCounter with the $^{210}$Pb ages derived from the upper part of core #2 that were used to constrain TC2016 (Fig. 6 and S7). These two independent dating methods are in good agreement down to 53 m (41 m w.e.; ~1940 CE), while they diverge thereafter. This offset can partly be explained by the fact that the initial

dating by $^{210}$Pb did not account for any layer thinning, statistically not evident from the data structure of measured activity concentrations. Additionally, a slight offset in the $^{210}$Pb ages due to imprecise determination of the background from supported $^{210}$Pb cannot be excluded. A plateau in activity concentration below a certain depth, clearly indicating the $^{210}$Pb background level (Gäggeler et al., 2020), was not observed and its determination was based on one single data point only. However, the extended visual annual layer counting from 53 m (41 m w.e.) to 57 m (45 m w.e.) connects well with the most recent

synchronization ties of the Alto dell'Ortles and SZ Pb records (Fig. S7). Therefore, annual layers were adopted to replace the $^{210}$Pb time markers initially used in TC2016.

**4 COPRA fitting of the dating horizons**

In line with the development of TC2016, we built a revised continuous depth-age relationship of the Alto dell'Ortles

cores (Fig. 6) by fitting all the empirical time markers (defined in meters of w.e. depth and years before 2012; Supplementary Table 2) within their linked uncertainty ranges by means of a Monte Carlo simulation (COPRA model, 2000 simulation runs). This provides a depth-age relationship with a linked depth-time dependent uncertainty (Breitenbach et al., 2012). However, in the most recent portion of the chronology (2011-1927), the original uncertainty of the single annual layers provided by StratiCounter (1–10 years) was retained and applied by linear interpolation to the depth-age relationship obtained using

COPRA.

Overall, 85 annual time markers provided by StratiCounter from 2011 to 1927 were combined with 22 tie points obtained from the match of the Ortles core #3 with the SZ Arctic Pb record (1907 CE-171 BCE) and five $^{14}$C time markers in the deepest part of the record (239 BCE back to 7000 years) (Fig. 6). As mentioned above, to bridge the time transition between the StratiCounter time markers and the Pb tie points, a visual pollen annual layer counting was performed between 1927 and





1900 CE (Fig. S7). However, we adopted in COPRA only the four oldest counted years (1903-1900 CE) that were averaged with the two most recent tie Pb ties (1909 and 1900 CE) to obtain a single constraint and its linked standard deviation. In this way the age uncertainty obtained by COPRA over the transitional interval should be more realistic. Likewise, in order to also obtain a smooth and robust transition between the two oldest periods constrained by the Pb tie points and the [14]C time markers

at around 200 BCE, we adopted two averaged values and their linked standard deviations obtained from two groups. Each consists of three time markers (402, 306, 442 CE and 232, 1, 71 BCE, respectively), allowing the consideration of the dating uncertainty and thus maximum resolvable differences in age, instead of an incorrect assumption of age reversal by COPRA for these sequences which are stratigraphically coherent, considering their uncertainties. Finally, the WIOC-14 [14]C dated sample at 1361 ± 204 CE, adopted in TC2016, was not used for input to COPRA because of its large uncertainty (Fig. 6c).

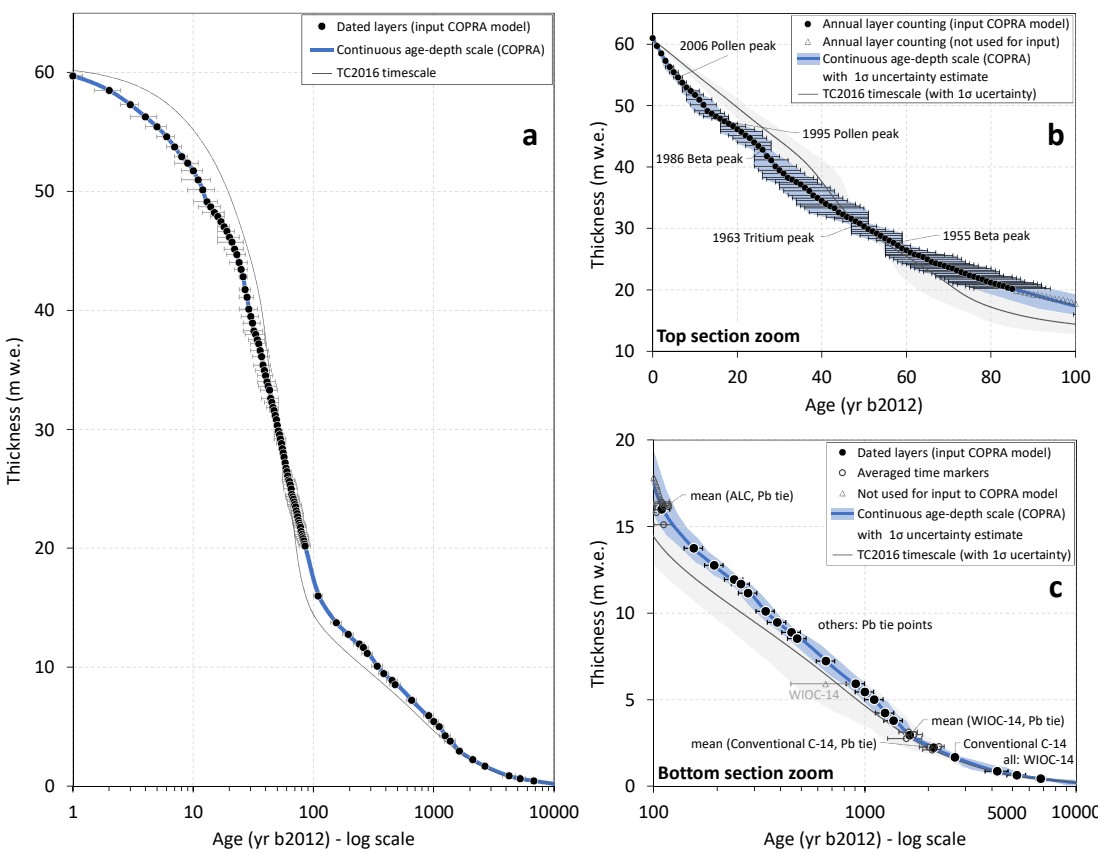

Figure 6: Comparison of the TC2016 and the revised Alto dell'Ortles timescales obtained by using COPRA and illustration of the different time markers used (see Table S2 for detailed description) over the entire time spanned by the Alto dell'Ortles ice core record (a), during the last 100 years (b) and in the older (>100 years) part of the record. Error bars indicate the dating uncertainty of the individual layers as determined by the linked dating methods (see text).



Comparison of the revised Alto dell'Ortles time scale and TC2016 is displayed in Fig. 6. The two timescales differ significantly during the most recent 30 years, corresponding to the upper firn portion of the core. As mentioned, in the revised time scale [210]Pb time markers were replaced by the annual layers obtained by StratiCounter (also taking in to account three additional time markers of 2006, 1995, 1986; Fig. 6 and S7). Besides the most recent time interval, the revised and the TC2016 chronologies essentially agree within their uncertainty (Fig. 6). However, the revised time scale is 100-300 years older between the last millennium and 7000 years ago (Fig. 6) while the overall uncertainty is significantly reduced during the last 2600 years (from ± 500 years or less in TC2016 to ± 100 years or less in this work).

## 5 Investigation of the empirical time scale by glaciological ice flow modelling

### 5.1 Model framework

We applied the Dansgaard-Johnsen (DJ) glaciological ice flow model (Dansgaard and Johnsen, 1969) to investigate the revised age-depth relationship at the Alto dell'Ortles drilling site. The DJ model (one-dimension; 1D), uses an approximation of the ice flow to calculate the vertical strain rate (thinning of annual ice layers with depth). It results from assuming the horizontal velocity ($v_x$) described by Glen's law, to be constant from the glacier surface down to a given height above bedrock (h; shear zone thickness), where it starts decreasing linearly with depth to zero at the bed (no-basal ice flow; i.e., ice frozen to bedrock; blue curve in Fig. 7). It should be noted, that the term h is the result of the DJ model-approximation for $v_x$ and subsequent mathematical reformulations, rather than being descriptive of an actually existing distinct separation between two zones.

A general limitation of modelling the age-depth relationship, independent on the complexity of the applied ice flow model, is the assumption of steady-state conditions. This is unavoidable unless additional glacier mass balance related information of the past is available. To understand our approach outlined below, it is important to recognize that under the assumption of steady-state at the drill site, net annual snow accumulation (in the accumulation zone equal to the mass balance at that point; Cogley et al., 2011) and $v_x$ along the glacier vertical profile are closely related. For reasons of mass conservation, if the amount of accumulated snow is not to increase the thickness of the glacier at that point, the same mass quantity must be removed from the ice column below. In more glaciological terms, the accumulated snow needs to be balanced by the horizontal strain and, due to the incompressibility of ice, the corresponding vertical strain rates (note the reduction to 1D, i.e., to the vertical axis). These fundamental principles are governed by the equation of continuity which, if integrated over the ice thickness, can be written to also relate to time and density (e.g., Whillans, 1977). In the DJ model, which considers these principles, the cumulative vertical strain of a layer at a certain depth over time (i.e., the total thinning) is related to the value of h. The DJ model accounts for the importance of density by using m w.e. as unit of length. This is essential, because in the firn section of the vertical column, a substantial part of the horizontal stress will not cause deformation (i.e. the thinning governed by the model related to the incompressibility of ice), but instead results in firn compaction in addition to the contribution from the overlying mass.



In conclusion, information about time (i.e. the age of a layer at a certain depth) can provide invaluable constraints for model parameters that are hardly accessible otherwise (equally for models of lower or higher complexity).

## 5.2 Determination of model parameters for the Ortles drilling site

For meaningful model output, the choice of parameter values is crucial. Here, we used two independent approaches for the determination of b and h: (A1) based on present-day glaciological observations (section 5.2.1), and (A2) based on the time information provided by the empirical dating described in this study (section 5.2.2). We further used parameters from A1 in a mixed approach (B) with A2 (section 5.2.3). In all cases, steady-state conditions were assumed, which for the applied model (DJ) can be defined as: i) constant glacier thickness (H), ii) no-basal ice flow, iii) constant annual net accumulation rates (b), and as a consequence, iv) a constant shear zone thickness (h). For the glacier thickness H and the condition of non-basal flow, we relied on present-day observations. For H, we adopted the present value of 74.88 m (61.15 m w.e.) (Gabrielli et al., 2016), implying that H has not changed significantly since glacier build-up at the drilling site ~7 kyrs BP. The "no-basal ice flow" assumption seems reasonable considering the ice temperature at bedrock which is still below 0°C (-2.8 °C in 2011; Gabrielli et al., 2012), despite the recent warming.

### 5.2.1 Approach A1: determination of b and h based on present-day glaciological observations

For b, the value of 1.0 m w.e. $y^{-1}$ as derived from field observations was considered as a best available estimate (Gabrielli et al., 2016; Festi et al., 2015). For the determination of h, we took advantage of the available data from englacial cumulative displacement measurements obtained with an inclinometer in Fall 2011 (borehole #2; see Fig. 4 in Gabrielli et al., 2016). The vector sum of the annual displacement along the two spatial axis (X, Y) yields a direct measure of $v_x$ for each depth (relative to the lowermost layer measured). Because the DJ model requires units to be in m w.e. (see section 5.1), we transformed the original displacement data (in m) to this unit using the smoothed density profile (Fig. 7), which accounts for spatial inhomogeneities (larger than the ice core cross section) in firnification. The profile indicates a clear imprint/variation related to the transition between the characteristic Ortles zones (Fig. 7): (i) the active layer, with a displacement signal observed at a thickness of ~52 m w.e., which is in agreement with determination based on englacial temperature measurements (personal communication Roberto Seppi), and (ii) the temperate zone, within 0.5 m of the depth previously reported (Gabrielli et al., 2016). Notably, the profile of horizontal displacement per year (i.e. $v_x$) allows determination of a best estimate value for the shear zone height h of 37.3 m w.e. (obtained by the method of least residual sum of squares between measured and approximated $v_x$). The degree of consistency between the DJ model approximation of the $v_x$ vertical profile with the actual measurements is remarkable (see Fig. 7). We consider the observed consistency as a strong argument for the applicability of the DJ ice flow model for the Alto dell'Ortles drilling site and likely also for other alpine glacier settings characterized by strong layer thinning near bedrock.



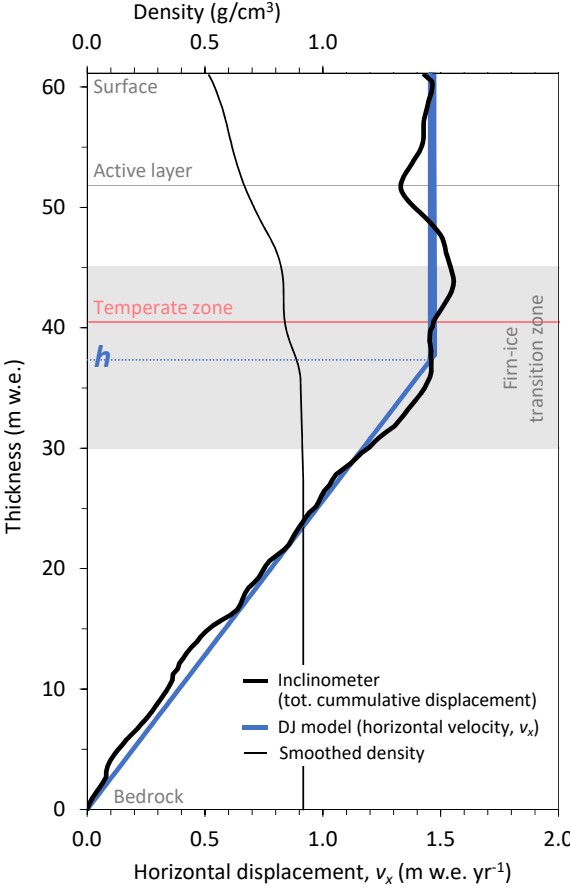

Figure 7: Smoothed vertical profiles of density and horizontal displacement (i.e., $v_x$) for the Alto dell'Ortles drilling site. Also indicated is the elevation above bedrock defined as shear zone thickness in the DJ model (parameter h), which was derived based on the displacement measurements (see main text). The firn-ice transition zone, typically 0.830 - 0.917 g/cm³, (Cuffey and Paterson, 2006) is indicated by the grey shading while the grey and red lines mark the bottom of the active layer and temperate zone, respectively (see main text).

**5.2.2 Approach A2: determination of b and h based on time information provided by the empirical dating**

In this case, both b and h were treated as free model parameters. The best set of values was then derived from minimization of the misfit between the DJ modelled ages and those of the empirically dated layers provided in Table S2 (least sum of time-weighted squares of residuals; Fahnenstock et al., 2001). This approach results in values of 1.05 m w.e. y⁻¹ and 36.1 m w.e. for b and h, respectively.



### 5.2.3 Approach B: combination of approaches A1 and A2

Either b or h was pre-set as derived in A1 while the respective other parameter was kept free, with its value then derived following the minimization methodology described for A2. By prescribing b = 1.0 m w.e. yr$^{-1}$, the value of h for least misfit is 32.6 m w.e. (approach Bb). By prescribing h = 37.3 m w.e., b results 1.06 m w.e. yr$^{-1}$ (approach Bh).

### 5.3 Ice flow model results, interpretations and limitations

The model parameter values, either inferred from present-day observations (A1) or based on information related to time (A2), lay within a relatively narrow range (1.0 - 1.06 m w.e yr$^{-1}$ and 32.6 - 37.3 m w.e. for b and h, respectively). This is remarkable considering that:

i.     for A1, b is based on only 8 years of observations while h presumably reflects flow conditions of a significantly longer time-span due to delayed glacier response to changing conditions, see e.g. Cuffey and Paterson, 2006),

    ii.    for A2, b and h reflect the respective average for a time-period of ~7000 years.

While b and h , as determined by A1 are due to glacier response times likely not directly related, A2 highlights how empirically derived time markers allow obtaining reasonable and related estimates of model parameters (even if recent glaciological

observations would be lacking). By constraining parameters b and h with time information (approaches A2, B), model results, fulfilling the equation of continuity and respecting mass conservation, in agreement with empirical dating can be obtained (see section 5.1). This is illustrated by the fact that the time span derived empirically between the lowermost dated layer and the surface (6437-7165 years, 1σ uncertainty range) is consistent with the model result using the sets of parameters from A2 (6730 years) and B (6630 and 6780 years for Bb and Bh, respectively). Here, B further provides insights about robustness and

sensitivity of A2, demonstrating how a difference in the estimate of a single model input value requires adjustment of at least one of the other model parameters, if the time span contained in the ice column is fixed.

The modelled timescales obtained by using the different sets of parameters are displayed in Fig. 8 (A1, A2 and Bb; Bh not shown as it is visually indistinguishable from A2 and Bb). The relatively close range of values derived for the respective parameters (b, h) and the agreement of the modelled timescales obtained from A1, A2 and B, suggest that over the entire time

period covered by the ice core, b (and accordingly h) on average must have been close to the modern observed values. While for A1, the modelled time-span contained in the ice column (7180 years) slightly exceeds the observations, the results for all sets of parameters have in common to agree well with the Alto dell'Ortles revised empirical time scale for the last ~100 years. Good agreement is also obtained for the oldest part of the record (around 7000-4000 years ago). However, for the intermediate part (4000 - 100 years ago) all approaches clearly fail to reproduce the observations, underestimating the age of the ice by

several decades to about one millennium in the upper and lower portion of this section, respectively (Fig. 8). This significant discrepancy suggests that during this time, conditions were different notably from the middle Holocene until the end of the Little Ice Age (LIA; 1250-1850 CE) (Pages K Consortium, 2013). The most likely explanation is a lower annual net accumulation during that time (also see below).





To investigate the hypothesis of changes in net annual accumulations rates over that time period, we used pre-set, hand-picked values for b and h for model input (approach C, Fig. 8). For b (0.3 m w.e. yr$^{-1}$) and h (15 m w.e.) at least a partial match of the DJ model with our revised empirical time scale can be obtained also for the intermediate section of the core (Fig. 8). Thus, while the steady-state assumption for modelling can be valuable in order to estimate the time span of the ice archive

5   (e.g. Lüthi and Funk, 2000) it seems invalid to derive an accurate age-depth relationship based on our results, at least for the Alto dell'Ortles drill site. Here we remark that the selected value of b = 0.3 m w.e. yr$^{-1}$ is not constrained in any way, and can thus be considered qualitative only, to illustrate a possible change in net accumulation for the discussed period. To derive a quantitative estimate of b, a non-steady-state approach under consideration of continuity would be required (inverse modelling, see below).

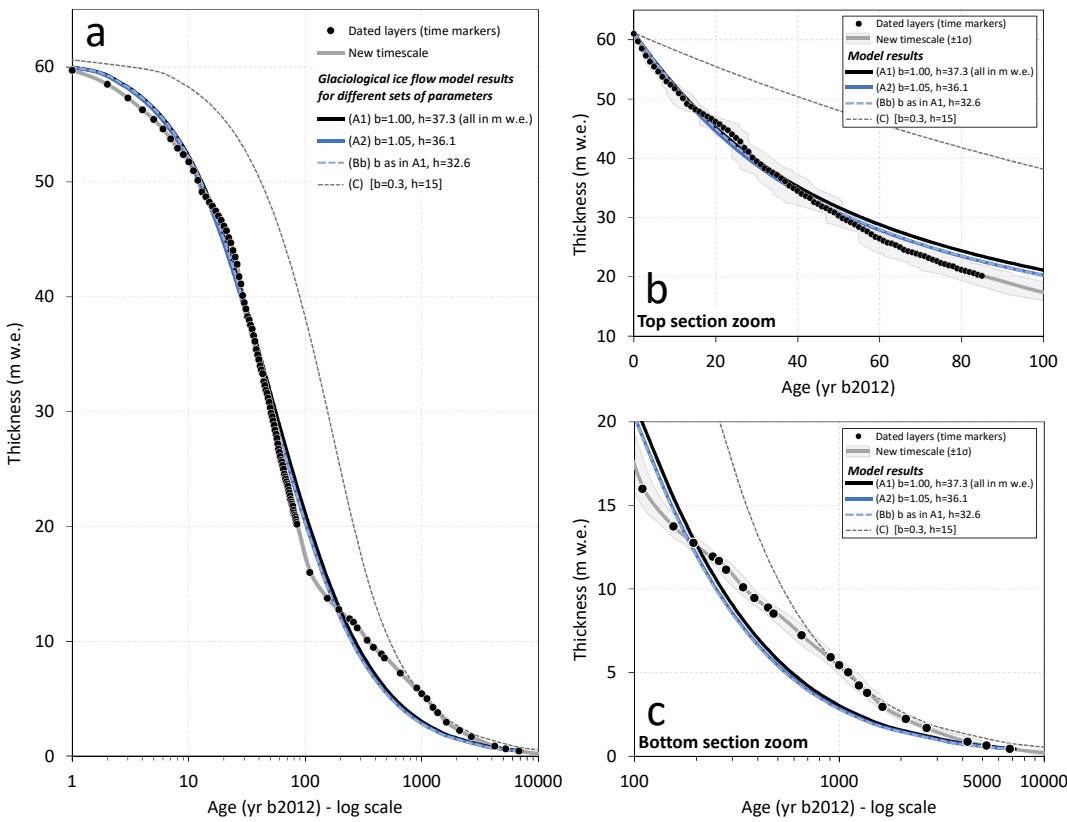

Figure 8: Comparison of the revised empirical timescale with several runs of the Dansgaard-Johnsen glaciological ice flow model for the indicated sets of parameter values (see text for details); (a) for the entire time-span covered by the Alto dell'Ortles ice core record, (b) the last 100 years, and (c) the part older than 100 years.





Our interpretation of the ice flow modelling that suggests lower snow accumulation during the central period of the Holocene (100-4000 years ago), particularly during the LIA, seems reasonable. A change in conditions for the depth interval corresponding the LIA, recorded right below ~57 m depth (or ~45 m w.e.), is consistent with intensification in visible ice layer thinning in the Ortles ice core cores (see fig 7b in Gabrielli et al., 2016) which is concomitant with a sudden increase in the frequency of the stable isotope water signal (Fig. 1). As different paleoclimate archives indicate higher precipitation in the Alps during LIA (Magny et al., 2011 and references therein), a lower snow accumulation on Alto dell'Ortles may be explained by a more efficient wind erosion of colder and light (low density) surface snow due to lower air temperatures, particularly during the warm seasons from late spring through fall. If modern temperatures are the warmest since the Northern Hemisphere Climate Optimum, it is possible that this hypothesis could hold from the beginning of the middle Holocene (4000 years ago) until the end of LIA (19[th] century).

In any event, our suggestion of variations in snow accumulation over time is not unprecedented for high altitude glaciers. In the few low latitude drilling sites where it was possible to measure changes in past snow accumulation by layer counting (Thompson et al., 2000; Winski et al., 2017) or based on [14]C ages; (Herren et al., 2013), these observations implied non-steady-state conditions and impacted the timescale. While inverse modelling approaches that consider non-steady-state conditions and take advantage of empirically derived time markers are well established for polar drilling sites (e.g. Buiron et al., 2011; Buchardt and Dahl-Jensen, 2008), efforts in this direction are also clearly needed for glaciers from high altitude/low latitude sites.

**6 Conclusions**

A revised chronology was obtained for the Alto dell'Ortles ice cores by incorporating an absolute time marker from a newly discovered [14]C dated macro-organic fragment (232 ± 126 BCE); by matching the non-crustal Pb flux record to a corresponding accurately (±5 years) dated polar ice core record from Severnaya Zemlya (Russian Arctic) between ~200 BCE and ~1900 CE; and by counting annual layers (from pollen, dust and $\delta^{18}$O records) visually and by means of an automatic algorithm (StratiCounter) between 1900 and 2011. All the time markers were combined by using the COPRA fitting model to obtain a more accurate chronology (from ± 5 up to ± 100 years) that is about 200 years older between the last millennium and 7000 years (still however within the larger uncertainty (± 500 years) of the preliminary time scale). An ice flow model investigation of this revised empirical chronology was performed by adopting a simple 1D Dansgaard-Johnsen model whose results suggest that non-steady state conditions (e.g. changes in snow accumulation rate over time) need to be considered to explain the age-depth relationship obtained. This concerns in particular the Little Ice Age (1250-1850 CE) when a decrease in snow accumulation rate was inferred. Clearly non-steady state approaches, currently used in polar regions, will need to be employed to model the ice flow also at high altitude-low latitude ice core drilling sites.

**Data availability:** The data presented in this work are archived at the National Oceanic and Atmospheric Administration World Data Center-A for Paleoclimatology:



**Author contribution:** PG and TJ designed the structure of the revised chronology and wrote the paper; MB, PG and CB performed the continuous flow analyses of Pb; DF, WK and KO performed the pollen analyses and their interpretation; MW and DF adopted the StratiCounter algorithm to count annual layers; GD and BS studied the stable isotopes data set; PG and MB performed the matching of the ice core records; TJ designed/run the ice flow model experiments and evaluated the results
with PG and MS. All the authors discussed and reviewed the manuscript.

**Acknowledgments**

This work is a contribution to the Ortles project, a program supported by NSF awards #1060115 and #1461422 to The Ohio State University (OSU) and by the Fire Protection and Civil Division; Südtirol, Abteilung Bildungsförderung,
Universität und Forschung of the Autonomous Province of Bolzano - Südtirol in collaboration with the Forest Division of the Autonomous Province of Bolzano - Südtirol and the National Park of Stelvio. We are grateful to: Giuliano Bertagna and Ping Nan Lin for performing Pb and δ¹⁸O analyses at OSU; John Bolzan, Ian Howat, Luca Carturan and Roberto Seppi for useful discussions linked to the modelling section; and Mary Davis for editing the manuscript. This is Ortles project publication 10 (www.ortles.org) and Byrd Polar and Climate Research Center contribution C1615.

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
