# Peer review of "Multimillennial synchronization of low and polar latitude ice cores by matching a time constrained Alpine record with an accurate Arctic chronology"

_Climate of the Past, 2022_

## Referee Comment (RC2)

Review of "Multimillennial synchronization of low and polar latitude ice cores by matching a time constrained Alpine record with an accurate Arctic chronology" by Paolo Gabrielli , Theo M. Jenk , Michele Bertó , Giuliano Dreossi , Daniela Festi , Werner Kofler ,Mai Winstrup , Klaus Oeggl , Margit Schwikowski , Barbara Stenni and Carlo Barbante

The manuscript of Gabrielli et al., presents a revised dating of three non-polar ice cores drilled in a distance of a few meters in 2011 at the glacier Alto dell'Ortles (3859 masl, Eastern Alps, Italy), and for which a first dating was published in 2016 (Gabrielli et al., 2016, denoted as TC2016 in the following)). To achieve a refined common chronology of the three cores, the depth scale matching of the tree cores is revisited on the basis of the 18O records of all three cores, and by adding additional depth match markers by using the Pb concentration profile from two of the three cores.

The absolute timescale of the common depth scale is then obtained by:
a) using the already existing chronology of 2016 which was based on 210Pb, tritium, beta activity and 14C determinations,
b) using additional time markers originating from:
- one 14C dating of an organic fragment giving a time marker at 232 ± 126 BCE,
- the synchronization of the excess (non-crustal) Pb flux record to the one of a well-dated Arctic ice core record (McConnell et al., 2018) from ~1900 CE to ~200 BCE,
- automated and visual counting of annual layers based on pollen and d18O and dust records from 2011 to 1900.
c) applying the obtained time markers in a continuous timescale on which the depth-age Monte Carlo based model COPRA (developed by Breitenbach et al., 2012).
Finally a 1D flow model (Dansgaard et al., 1969) was used to test the revised depth age relation and to test the hypothesis of steady-state conditions of the glacier side with a 1D flow model.

This study addresses the for ice-core archives very important scientific question of the ice core depth age relation, which is in the scope of CP. The study presents new data and the manuscript is structured adequately with respect to the aim of the manuscript. In my opinion the manuscript may be suitable for publication after major revisions were made.

One of the major concerns I have, is the use of the 14C data within the revised dating attempt, since at least two recent dating studies of high alpine ice cores showed that it cannot be necessarily assumed that the depth-age chronology of small scaled alpine ice cores is free of any discontinuities or age reversals (see e.g. Hoffmann et al., 2017, Preunkert et al. 2019).

Having that in mind, a critical look on the 14C raw date published within the first dating attempt (TC2016), one becomes aware that the Ortles chronologies are not free of doubt of

a discontinuity in the depth-age scale, although glaciological investigations did not show any hints on such an occurrence (see TC2016). Note that this was neither the case in the study of Hoffman et al. 2017 at Colle Gnifetti (4450 m asl Swiss Alps) neither within the study of Preunkert et al., 2019) at Col du Dome (4250 m asl French Alps). Therefore, a potential occurrence of such a feature should be addressed in the revised dating attempt.

I took the liberty to draw a graph (see Figure 1 below) in which the available non averaged 14C results from TC2016 Table 2 (see also below) and the 14C result of this manuscript are reported over the depth of core 3. To do so the depth scale of core 1 between 68 and 72 m depth was roughly matched core 3 by applying: depth core3 = depth core1 - 1m.

As could be seen there are in core 1 as well as in core 3 independent signs of a depth-age disturbance around 71m depth (equivalent of core 3).

Within the TC2016 dating attempt the age reversal in section 102 of core 1 was eliminated using the mean age of the three subsamples of section 102, the subsection 103 sample of core 1 was not used since it would have represented a chronological inversion, and it had a increased risk of contamination during sample preparation.

As mentioned above since the time of the publication of the first dating attempt in 2016, it became obvious that one can not expect a continuous undisturbed depth-age relation at non alpine glacier sites. Therefore, either additional 14C analysis should be undertaken at the depth between 70 and 72m (depth equivalent of core 3) to confirm or exclude a depth age disturbance, or address this point in this manuscript and discuss the consequences of a potential disturbance of the continuity in the depth age profile around 71 m depth (equivalent of core 3) in detail throughout the manuscript (i.e. within the comparison of the non-crustal Pb Ortles data with the Arctic Pb ice core data (section 3.2), and within the application of the COPRA model (section 4) and the Dansgaard-Johnsen model (section 5).

Table 2 from TC2016:

**Table 2.** $^{14}$C analyses of the particle organic fraction (WIOC) obtained from the four sections (tubes) of the Mt. Ortles ice cores no. 1 and no. 3. Except for Sect. 103b, the samples were analysed in three subsamples (top, middle, bottom). $^{14}$C determination in Sect. 105b (core no. 1) refers to a larch leaf that was found in the ice. Samples reported in bold are those also included in Table 3. Note the notation used for calibrated ages in yrs b2012.

| Core no. | Tube no. | Measure | Top depth (m) | Bottom depth (m) | WIOC (µg) | F$^{14}$C | $^{14}$C age (yrs BP) | Cal age (yrs cal BP) | µcal age (yrs cal BP) | µcal age (yrs b2012) | σ (years) |
|---|---|---|---|---|---|---|---|---|---|---|---|
| 1 | 98b | WIOC | 68.26 | 68.49 | 17.11 | 0.971 ± 0.024 | 236 ± 199 | (−4–461) | 279 | 341 | 167 |
| 1 | 98b | WIOC | 68.49 | 68.73 | 17.86 | 0.911 ± 0.021 | 749 ± 185 | (550–902) | 732 | 794 | 163 |
| 1 | 98b | WIOC | 68.73 | 68.96 | 15.06 | 0.900 ± 0.024 | 846 ± 214 | (562–974) | 824 | 886 | 192 |
| | | 98b WIOC$^b$ | 68.26 | 68.96 | | **0.927 ± 0.025** | **609 ± 217** | **(331–790)** | **595** | **657** | **205** |
| 3 | 102 | WIOC | 70.87 | 71.14 | 7.98 | 0.784 ± 0.043 | 1955 ± 451 | (1395–2431) | 2011 | 2073 | 517 |
| 3 | 102 | WIOC | 71.14 | 71.35 | 7.15 | 0.867 ± 0.065 | 1146 ± 602 | (537–1720) | 1253 | 1315 | 620 |
| 3 | 102 | WIOC | 71.35 | 71.57 | 13.28 | 0.818 ± 0.032 | 1614 ± 314 | (1187–1921) | 1590 | 1652 | 347 |
| | | 102 WIOC$^b$ | 70.87 | 71.57 | | **0.823 ± 0.027** | **1565 ± 264** | **(1262–1818)** | **1521** | **1583** | **286** |
| 1 | 103b | WIOC | 71.8 | 72.48 | 10.37 | 0.932 ± 0.037 | 569 ± 320 | (156–903) | 517 | 579 | 291 |
| 1 | 105b | Larch leaf | 73.25 | 73.25 | 68$^a$ | **0.728 ± 0.006** | **2550 ± 65** | **(2500–2752)** | **2612** | **2674** | **101** |
| 3 | 106 | WIOC | 73.73 | 74.02 | 10.91 | **0.628 ± 0.031** | **3737 ± 397** | **(3593–4787)** | **4173** | **4235** | **523** |
| 3 | 106 | WIOC | 74.02 | 74.24 | 11.50 | **0.568 ± 0.030** | **4544 ± 424** | **(4623–5715)** | **5178** | **5240** | **530** |
| 3 | 106 | WIOC | 74.24 | 74.47 | 18.47 | **0.481 ± 0.020** | **5879 ± 334** | **(6354–7156)** | **6742** | **6804** | **365** |

$^a$ Pure C extracted after combustion.
$^b$ Combined values from the three subsamples of tubes 98b and 102.

[Figure]

Figure 1: Non averaged 14C results from TC2016 Table 2 (see above) and the 14C sample of this manuscript over the depth of core 3. Blue points correspond to 14C samples from core 1 which are reported on the depth of core 3. To do so the depth scale of core 1 between 68 and 72 m depth was roughly matched to the one of core 3 by applying: depth core3 = depth core1 - 1m as read from Figure 2 of the manuscript.

Another major concern I have is the traceability of the proceeding of depth and time scale matching using d18O and Pb profiles, i.e. for the depth scale matching of the three cores and dating of the common Ortles depth scale between 1907 CE and ~200 BCE by matching the non-crustal Pb Ortles profile with the Pb data of a well dated Arctic ice core.

Although the authors describe their proceeding very detailed, it is not easy to follow their proceeding and arguments.

To improve that:

1. A table or a scheme should be added in section 2, which gives an overview on: which parameters are available in which parts of which of the three cores, and their respective use in view of the dating procedure. Since different reference depth scales are used in the Figures (Fig 1. reference depth from core 2, Fig 2 reference depth of core 3, Fig 5 reference depth of core 1) that will be an important guidance for the reader throughout the different sections of the manuscript. In addition, a SI table would be very helpful which reports the used tie point depths of the three cores in absolute and water equivalent depths for each core.

2. Figure 1 is as it stands not useful for the reader since the matching of d18O profiles is not traceable. It would be better to use an illustration type as used in Fig. 2 means keep the individual depth scales and highlight tie point and their connections between the cores. Hereby prescind the principal tie points used to anchor the depth scale matching and their connections between the cores, vice versa the refinement tie points set afterwards. In addition, the y axis should be extended, in order that the variations of the d18O profiles become more obvious, especially in the lowest part of the cores.

3. Figures 2, 4, S2, S3, S4 should be changed. As it stands, the reliability of the depth scale matching via the comparison of the non-crustal Pb profile cannot be assessed by the reader since in the above-mentioned illustrations the non-crustal Pb data were logarithmized or logarithmic axes are used to present the Pb data. In addition the Pb and the non-crustal Pb data are not made available for the reader (at least I did not find them). Therefore, to make the reasoning of the Pb profile matchings of the three cores and between Ortles and the Arctic core traceable for the reader, no logarithmic data and y axis scales should be used in Fig4, Fig S2, S3, S4.

Among other things, this would allow:

- to see clearer at which point the two Ortles Pb records agree or differ between core 1 and core 3. Differences seem to be up to half a log scale and over up to one m depth which makes up several hundred years at this depth (e.g. around 65, 68 and 70 m depth of core 3). Since for the moment, only one of the profiles is chosen for the age matching with the Arctic ice core, this is an important point to be discussed.

- to give the reader the opportunity to see the quality and reliability of the matching between the Ortles and the Arctic (and the CG) non-crustal Pb profiles. E.g. the authors state in line 1 of page 10 "In general, the match between Alto dell'Ortles and SZ is excellent, except during 1650-1900 CE ..." however Fig 4, Fig.S2, Fig. S3 and suggest that the matching is rather arbitrary especially after ~ 600 CE the course over time of the Ortles non-crustal Pb and Arctic Pb differs significantly. As it stands (logarithmic scale) I am not convinced that the Pb matching with the Arctic core is reliable enough to shift the age of the 14C data to the limit of their 68% probability, and to neglect the 14C sample at 1361 ± 204 CE in section 4. It will be also very important to see, whether the assumption of a depth age disturbance would improve the Pb matching agreement of Ortles and the Arctic ice core or not.

An alternative to showing the Pb and non-crustal Pb data on non-logarithmic scales and making them public within this manuscript, might be to join the refined dating addressed here to the aimed environmental discussion of the Pb data in a common manuscript.

Other comments:
The application of the Dansgaard-Johnsen 1D model (see Dansgaard et al., 1996) on the revised depth-age relationship at the end of the manuscript (section 5) should be shortened. Only synthesized essential information of the model, the different input parameter, the outcome and the discussion of the results stay in the manuscript. Model description, input parameter determination of glaciological observations should be put in the SI.

For the comparison of non-crustal Pb Ti is used at CG and Rb at Ortles (Figures S2, S3, S4). Why Ti was not used as well at Ortles for consistency? Please report in the text the crustal values used for the Figures. Also discuss and provide errors in calculating non-crustal Pb. How large are the changes of Rb with age in the Ortles ice (please report the Rb profile in the Supplementary material).
* * *
References which are not cited in the manuscript but which should:

Hoffmann, H., Preunkert, S., Legrand, M., Leinfelder, D., Bohleber, P., Friedrich, R., & Wagenbach, D. (2018). A new sample preparation system for micro-14C dating of glacier ice with a first application to a high alpine ice core from Colle Gnifetti (Switzerland). Radiocarbon, 60(02), 517–533. https://doi.org/10.1017/rdc.2017.99.

Preunkert, S., J.R. McConnell, H. Hoffmann, M. Legrand, A. Wilson, S. Eckhardt, A. Stohl, N. Chellman, M. Arienzo, & R. Friedrich (2019) Lead and antimony in basal ice from Col du Dome (French Alps) dated with radiocarbon: A record of pollution during antiquity, Geophys Res Lett, doi:10.1029/2019GL082641.

---

## Author Comment (AC1)

*Reviewer 1*

We would like to thank Joe McConnel for reviewing our paper and for providing constructive suggestions. Please find our responses in the following:

*Does the paper address relevant scientific questions within the scope of CP? Yes.*
*Does the paper present novel concepts, ideas, tools, or data? Yes, but the approach is not as novel as implied in the manuscript.*

Corrected (see below for our answer to your specific comment 1)

*Are substantial conclusions reached? Yes, although the focus of this manuscript is only on establishing a new core chronology for the Alto dell'Ortles cores. The interpretation of the new Alpine lead record is reserved for a subsequent manuscript.*

*Are the scientific methods and assumptions valid and clearly outlined? Yes.*

*Are the results sufficient to support the interpretations and conclusions? More or less. The main advance in dating is the use of a relatively new technique where a poorly dated lead record (in this case from Alto dell'Ortles) is synchronized to a well-dated lead record (in this case a previously published record from the Russian Arctic). The wiggle matching between the records is somewhat arbitrary, however, and difficult to assess objectively. Here the authors use radiocarbon dating to assess the wiggle matching but the radiocarbon dates are too few and have too large uncertainties to allow a quantitative assessment of the wiggle matching. I don't mean to be too critical since the problem of quantitatively assessing wiggle matching is not unique to this study, but the subjective nature of the tie points should be openly acknowledged.*

We agree with the referee and this point is now openly acknowledged within then text. We also note that the subjective nature of the wiggle matching is already qualitatively and quantitively acknowledged within the text of the paper where a conservative uncertainty of the timescale was defined over the period of interest:

"*Wriggle-matching of records is inherently somewhat subjective….To assign a direct transfer of the dating uncertainty of the AN Pb record (±5 years) (McConnell et al., 2019) would underestimate the residual age discrepancy (204 and 288 years, see above) with the two WIOC-14C dated samples given the additional uncertainty associated with the synchronization procedure. As a consequence, an uncertainty of 10% of the matched age (expressed in years before 2012) ranging from 10 up to 200 years was assigned over this time interval. Ten years is a lower limit in the same order of the time uncertainty of the deepest sections dated by using annual layers down to 57 m depth in core #1, see below; 200 years is an upper limit in the same order of the mentioned discrepancy with the two WIOC-14C dated deeper samples*".

*Is the description of experiments and calculations sufficiently complete and precise to allow their reproduction by fellow scientists (traceability of results)? Yes.*

*Do the authors give proper credit to related work and clearly indicate their own new/original contribution? Improvements suggested including additional citations and acknowledgement of prior similar work.*

Changed accordingly (see below for our answer to your specific comment 1)

*Does the title clearly reflect the contents of the paper? Yes, although it describes only one component of what was reported.*

We have now revised the title to read "A multimillennial mid-latitude ice core chronology by synchronization with a polar Pb record combined with other empirical dating methods"

*Does the abstract provide a concise and complete summary? Yes.*

*Is the overall presentation well structured and clear? Fairly clear. The text could be improved for readability and shortened.*

We have now revised the readability and shortened the text by transferring some portions to two new sections of the supplementary information (see below).

*Is the language fluent and precise? For the most part, yes. The text in the Supplement could be edited to be more understandable in English.*

This section has been carefully revised by a native English speaker.

*Are mathematical formulae, symbols, abbreviations, and units correctly defined and used? Yes.*

*Should any parts of the paper (text, formulae, figures, tables) be clarified, reduced, combined, or eliminated? Some of the figures are primitive by today's standards and could be improved. For example, I found the comparison of the synchronized Russian Arctic and Alto dell'Ortles lead records in the Supplement (Fig. 2 bottom) to more compelling than the same presentation in Fig. 4 (bottom).*

We have now revised Figure 4, as well as Fig. 1, Fig S2, S3 and S4, accordingly to this comment.

*Are the number and quality of references appropriate? Yes, but important corrections and additional references are needed.*

Corrected and added (see below for our answer to your specific comment 1).

*Is the amount and quality of supplementary material appropriate? Yes.*

*Overview*
*Development of well-dated historical records of human impacts and climate are important for a broad range of disciplines, including the natural, physical, and social sciences, as well as the humanities. Polar and alpine ice cores record provide direct, often highly resolved records of past atmospheric and precipitation chemistry that reflect both natural and anthropogenic emissions. The caveat is that they are most useful only if the records can be properly dated. This manuscript describes the use of standard and non-standard dating techniques to develop a new chronology for the Alto dell'Ortles cores from the Italian Alps. As is typical of cores from relatively thin alpine glaciers, annual layer counting is used in the upper section with constraints provided by known time horizons (e.g., fallout from 1950s and 1960s atmospheric thermonuclear testing, as well as fallout from volcanic eruptions or other highly unusual but well-dated events such as major forest fires or Saharan dust events). In deeper sections of such cores, annual layer counting generally is not possible because of extreme flow thinning and strain so other techniques are needed for establishing the ice age. Here the authors used measurements of lead concentrations in two of the four (or maybe three) Alto dell'Ortles cores collected in 2011 to synchronize the deeper sections to a well-dated, previously published lead record from the Russian Arctic. Additional constraints were provided by radiocarbon dates of both water insoluble organic carbon and discrete organic material from the basal section of one of the cores. The lead synchronization technique used here is relatively new and this is the first effort that I'm aware of to use annual layer counting based on seasonal pollen variations and extreme pollen events as specific time markers.z While I support publication and am especially enthusiastic about the better-dated records of climate and human impacts in Europe that will result from the improved chronology, there are number of important issues that need to be resolved first.*

*Specific Comments*
*(1) In the abstract, the authors describe as "novel" the approaches used to redate the cores. The most significant relatively new part of their approach is lead-based synchronization to a well-dated polar ice core which also is the manuscript title. In fact, exactly this lead synchronization approach has been used previously to date rapidly*
*thinning ice cores over the Common Era and beyond. Specifically, Osman et al., (2021) used this technique of annual layer counting in the upper section and lead synchronization in the lower section on a coastal dome core from Greenland. Similarly, Preunkert et al. (2019) used this techniques on a core from the French Alps, including additional constraints from radiocarbon dating of organic material in the deep ice corresponding to antiquity. Therefore, "novel" should be removed from the abstract and these earlier applications at least mentioned and briefly discussed to provide context for the current study. Citations to these earlier publications obviously should be added as well.*
*Preunkert, S., J.R. McConnell, H. Hoffmann, M. Legrand, A. Wilson, S. Eckhardt, A. Stohl, N. Chellman, M. Arienzo, & R. Friedrich (2019) Lead and antimony in basal ice from Col du Dome (French Alps) dated with radiocarbon: A record of pollution during antiquity, Geophys Res Lett, doi:10.1029/2019GL082641.*

*Osman, M., B.E. Smith, L.D. Trusel, S.B. Das, J.R. McConnell, N. Chellman, M. Arienzo, & H. Sodemann (2021)Abrupt Common Era hydroclimate shifts drive west Greenland ice cap change, Nature Geoscience, doi:10.1038/s41561-021-00818-w.*

We have now removed the word "novel" from the abstract. Within the text we also provide additional context on the use of Pb as a tool for matching, including the two suggested references.

*(2) The citation for the published Russian Arctic lead record is incorrect. It should be McConnell et al., 2019, not McConnell et al., 2018 (these are different publications and not simply the result of typos in their text).*
*McConnell, J.R., N.J. Chellman, A.I. Wilson, A. Stohl, M.M. Arienzo, S. Eckhardt, D. Fritzsche, S. Kipfstuhl, T. Opel, P.F. Place, & J.P. Steffensen(2019) Pervasive Arctic lead pollution suggests substantial growth in Medieval silver production modulated by plague, climate and conflict, Proc Natl Acad Sci U.S.A., doi:10.1073/pnas.1904515116.*

Corrected. Thank you for spotting this.

*(3) The use of SZ (the island of Severnaya Zemlya) rather than AN (the ice cap Akademii Nauk which is one of several glaciers/ice caps on SZ) as the ice core name for the Russian Arctic lead record is somewhat confusing. This may be because some earlier publications from the German/Russian team that collected and first analyzed the core referred to it both as SZ (e.g., Fritzsche et al., Annals of Glaciology, 2006) and AN (e.g., Opel et al., Journal of Glaciology, 2009; Opel et al., Climate of the Past, 2012). However, the lead record used here was published as the AN record in McConnell et al., 2019 and so the core name AN should be used here as well.*

We agree. Changed from SZ to AN in the manuscript.

*(4) It appears from Fig. 2 (bottom graph) that there are sometimes very large differences (nearly an order of magnitude for some periods) in the lead concentrations measured in the two Alto dell'Ortles cores. These aren't just short term differences but decadal or longer differences that I find is quite unexpected in two nearby cores. Please elaborate. Do these differences mean that the lead fluxes measured in these Alpine cores are not regionally or even locally representative?*

Different Pb concentrations in cores #1 and #3 are just due to the different acid leaching time between continuous flow analyses (CFA; using online acidification; core #3) and the discrete analyses (adopting acidification of aliquots for sample preparation; core #1) as illustrated in various papers (e.g. Arienzo et al. EST 2019, Uglietti et al AG, 2014., Rhodes at al.  CG 2011). Differences are larger at low Pb concentration levels probably because these are mostly characterized by crustal Pb that, unlike anthropogenic Pb, is less acid leachable (Arienzo et al. EST 2019). Nevertheless, different acidification methods do not affect trends and the Pb features (maxima, minima, fingerprint variations) used for wiggle matching. Further, when finally selecting the temporal tie points with the AN Arctic core, only features that were

reproduced in both Ortles Pb records were considered. This note has been summarized and added to the manuscript.

*In addition, the tie point at ~69.5 m between the two Alto dell'Ortles lead records is incorrect – or at least not optimal. Correcting it would improve the agreement between the lead records and so make the chronologies more consistent.*

This minor offset is a consequence of matching the high resolution CFA Pb record (about 0.2 cm, in Ortles core #3) with the lower resolution Pb record based on discrete samples (about 4 cm, in Ortles core #1). In general these offsets are not noticeable. In this exceptional case, a large change in Pb concentrations occurs quickly at that depth (69.5 m) where a strong thinning of the ice layers occurs. This minor discrepancy (up to 4 cm or about 5 years at this depth) is negligible as it is well within the age uncertainty adopted within this time interval (10% of the age or 100 years at this depth). We therefore consider this of negligible importance. We have added a sentence about the limitation caused by non-identical sampling resolution to the manuscript, pointing out that any potential bias is significantly smaller than the final dating uncertainty reported.

*(5) The new age scale is quite different from TC2016 even in the upper 100 m where both chronologies presumably are based largely on annual layer counting (albeit with constraints). This seems quite surprising. I understand that the new chronology incorporated pollen records in the upper part of the core but what caused the annual layer counting in the original chronology to be so far off? Presumably TC2016 was based on the same δ18O and dust measurements as in the current study. Please elaborate.*

The reviewer probably missed the point that the upper part of the initial TC2016 timescale was not based on any annual layer counting but was constrained by the 3H peak, Beta emissions and 210Pb dating only. The discrepancy observed is already discussed within the main text in paragraph 3.3 and also illustrated in supplementary info Fig S7.

*(6) I find the use a logarithmic age axis in the flow modeling sections (Fig. 6 showing annual layer thickness vs age) rather confusing. Why did you use a logarithmic scale? The main point of the manuscript is the redating of the deeper core (below 100 m) so shouldn't that be emphasized rather than the top 100 m?*

This probably relates to comment 5 above as the dating of the upper part was indeed revised and, when compared to TC2016, now also includes many more additional age constraints (e.g. pollen peaks are used to count annual layers). Log and non-log scales are adopted to provide an overall picture (panel a) and to allow visibility of details in the zoomed sections which would not be visible otherwise. We thus prefer to keep the figure as is, and believe that with the selected axis description, specifically pointing to the log-scale, any potential confusion by the reader will immediately be resolved.

*(7) In the third paragraph of the introduction, you say that four cores were collected from the Alto dell'Ortles site in 2011. After that, however, I find only a discussion of three cores. Did I miss something?*

Only the longest Mt. Ortles cores #1, #2 and #3 (they are all about 75 m long) were analyzed while a 4[th] shorter core (about 60 m; bedrock was not reached) was not analyzed and it was designated to be preserved for future analysis. We now clarify this within the text in the introduction.

*(8) I don't find Fig. 1 particularly compelling or informative. Is the point to show that the water isotopes are in better agreement once the new lead-based and other improvements in the tie points between cores are made? If so, it would be much clearer to show this by overplotting the original and improved water isotope records or by using cross plots. Improvements could be quantified by showing how the correlations between different core records have improved either overall or for specific depth/time sections.*

Fig. 1 has now been revised and much more information is provided including the comparison of the stable isotopes records before and after the matching. We now report the linear correlation coefficients (r) of the matched stable isotope records and their levels of significance (always p < 0.01) within the text in order to quantify the improvement of the alignments. In TC2016, 17 tie points between core #2 and # 1 (linear correlation r = 0.72) and 14 tie points between core #2 and #3 (r = 0.67) were used. In the revised chronology 122 tie points between core #2 and #1 (r = 0.78) and 87 tie points between core #2 and #3 (r = 0.79) were adopted. Overall, the moderate increase in r suggests that Fig 1 illustrates a subtle refinement of the depth alignments performed in the 2016TC.

Correlation of the two stable isotope records in core #2 and #3 before and after the use of Pb ties is essentially identical (r=0.79) because of the already highlighted low resolution of the stable isotope record in core #3. In this revised chronology, the major improvement in aligning the depths is linked to the depth intervals (Fig. 1 red tie points) linked by using the Pb ties from cores #1 and #3 where the two Pb records show a final correlation of r = 0.91 (p < 0.01). This is now mentioned within the text.

---

## Author Comment (AC2)

Reviewer 2

We thank Reviewer 2 for the review of our paper and for providing constructive suggestions. Please find our responses in the following:

*The manuscript of Gabrielli et al., presents a revised dating of three non-polar ice cores drilled in a distance of a few meters in 2011 at the glacier Alto dell'Ortles (3859 masl, Eastern Alps, Italy), and for which a first dating was published in 2016 (Gabrielli et al., 2016, denoted as TC2016 in the following)). To achieve a refined common chronology of the three cores, the depth scale matching of the tree cores is revisited on the basis of the 18O records of all three cores, and by adding additional depth match markers by using the Pb concentration profile from two of the three cores.*

*The absolute timescale of the common depth scale is then obtained by:*

*a) using the already existing chronology of 2016 which was based on 210Pb, tritium, beta activity and 14C determinations,*

*b) using additional time markers originating from:*

*- one 14C dating of an organic fragment giving a time marker at 232 ± 126 BCE,*

*- the synchronization of the excess (non-crustal) Pb flux record to the one of a well-dated Arctic ice core record (McConnell et al., 2018) from ~1900 CE to ~200 BCE,*

*- automated and visual counting of annual layers based on pollen and d18O and dust records from 2011 to 1900.*

*c) applying the obtained time markers in a continuous timescale on which the depth-age Monte Carlo based model COPRA (developed by Breitenbach et al., 2012).*

*Finally a 1D flow model (Dansgaard et al., 1969) was used to test the revised depth age relation and to test the hypothesis of steady-state conditions of the glacier side with a 1D flow model.*

*This study addresses the for ice-core archives very important scientific question of the ice core depth age relation, which is in the scope of CP. The study presents new data and the manuscript is structured adequately with respect to the aim of the manuscript. In my opinion the manuscript may be suitable for publication after major revisions were made.*

*One of the major concerns I have, is the use of the 14C data within the revised dating attempt, since at least two recent dating studies of high alpine ice cores showed that it cannot be necessarily assumed that the depth-age chronology of small scaled alpine ice cores is free of any discontinuities or age reversals (see e.g. Hoffmann et al., 2017, Preunkert et al. 2019).*

*Having that in mind, a critical look on the 14C raw date published within the first dating attempt (TC2016), one becomes aware that the Ortles chronologies are not free of doubt of a discontinuity in the depth-age scale, although glaciological investigations did not show any hints on such an occurrence (see TC2016). Note that this was neither the case in the study of Hoffman et al. 2017 at Colle Gnifetti (4450 m asl Swiss Alps) neither within the study of Preunkert et al., 2019) at Col du Dome (4250 m asl French Alps). Therefore, a potential occurrence of such a feature should be addressed in the revised dating attempt.*

We agree with the reviewer that per-se "it cannot be necessarily assumed that the depth-age chronology of small scaled alpine ice cores is free of any discontinuities or age reversals".  We also observe that this is an already widely accepted statement particularly when dating ice cores close to bedrock.

We are aware that, based on 14C dating, indications for ice folding were reported in the study of Hoffman et al. 2017 at Colle Gnifetti and of Preunkert et al., 2019 at Col du Dome. However, we would like to note that the data-set and data structure in the two studies cited are different from what is presented in our manuscript (and in TC2016). In Preunkert et al., 2019, the possibility of an age reversal is mentioned, based on a single data point for which a slight contamination could not be excluded (the potential reversal is disregarded later in the paper for the reconstructed pollution record). In Hoffman et al. 2017, the potential reversal does not seem to rely on one single data point only. However, the data over the section in question is generally rather noisy making a firm conclusion difficult based on statistics (data scatter, also see e.g. Fig 8 in Liciulli et al., 2020). In addition, in the case of the Colle Gnifetti drilling site, 14C dating of another core does not suggest any age reversal (Jenk et al., 2009, Sigl et al., 2009), which is in contrast with the result of Hoffmann et al.,2017.

Remarkably, as the referee also points out, our glaciological investigations illustrated in TC2016 do not show any hint of such an occurrence and no indication for ice folding was observed. We note this is reinforced by the excellent match of the Pb Ortles and Russian Arctic records at that depth. Below we also show that an indication of disturbance/age reversal is unlikely to be inferred from our 14C data.

*I took the liberty to draw a graph (see Figure 1 below) in which the available non averaged 14C results from TC2016 Table 2 (see also below) and the 14C result of this manuscript are reported over the depth of core 3. To do so the depth scale of core 1 between 68 and 72 m depth was roughly matched core 3 by applying: depth core3 = depth core1 - 1m.*

*As could be seen there are in core 1 as well as in core 3 independent signs of a depth-age disturbance around 71m depth (equivalent of core 3).*

*Within the TC2016 dating attempt the age reversal in section 102 of core 1 was eliminated using the mean age of the three subsamples of section 102, the subsection 103 sample of core 1 was not used since it would have represented a chronological inversion, and it had a increased risk of contamination during sample preparation.*

We thank the reviewer for the effort to compile and plot this data. We note that the three WIOC 14C measurements that we averaged at 71 m depth (orange points in the graph prepared by the Reviewer) show what is technically considered a "resolvable reversal" (Breitenbach et al., 2012) where a stratigraphic coherent temporal solution can always be found within the provided error bars. Thus, from a statistical point of view, these three points cannot be used as evidence of a reversal. In addition, based on the final age scale, the depth interval of these three subsamples is less than 1 m, containing around 200 years. It is clear that an age reversal in this section cannot be resolved, considering the achieved 14C dating uncertainty of > 200 years. Based on these observations, we have high confidence in the mean age of the three combined  samples, at least within the uncertainty assigned.

Concerning the #103 sample (in blue at 71 m depth in the graph prepared by the Reviewer), this was indeed not adopted in TC2016 (as well as in this revised chronology) where for this specific sample we reported in our previous publication that *"the amount of filtered WIOC from the first subsample* (out of the three composing this sample) *was estimated to be insufficient, and therefore the three subsamples were filtered together, resulting in a total ice volume exceeding our standard dimensions, possibly introducing a larger blank because of the modified treatment (i.e. increased potential for contamination due to a higher number of steps during sample processing)".* In other words this particular sample was most likely affected by small variations in the corresponding process blank. Thus this single data point is most likely indicative of a single outlier whose occurrence outside the 2 sigma range can however be statistically expected in a set of 10 samples. Thus sample #103 also cannot provide evidence of an age reversal.

These two observations are strongly supported by the compelling match of this ice section with the Arctic Pb record at that depth /time (please see the combined info in Figures 2 (~71 m depth) and 4 (~350 CE age). This comment has been summarized within the text.

*As mentioned above since the time of the publication of the first dating attempt in 2016, it became obvious that one cannot expect a continuous undisturbed depth-age relation at non alpine glacier sites.*

Please see replies above.

*Therefore, either additional 14C analysis should be undertaken at the depth between 70 and 72m (depth equivalent of core 3) to confirm or exclude a depth age disturbance,*

We agree that additional 14C measurements over that 2 m depth section could add additional evidence. While we are confident this would not change our conclusions (see replies above) unfortunately no such additional measurements are possible with the available ice left from Mt. Ortles, as relatively large amounts of ice mass are required for the 14C dating (for the Mt Ortles ice around 400-600 g of ice per 14C measurement to ensure sufficient mass of carbon > 10 μgC, for the analysis by gas-ion source AMS) that is unfortunately not available in any of the cores at that depth that was already intensively sampled for 14C and other analyses.

*or address this point in this manuscript and discuss the consequences of a potential disturbance of the continuity in the depth age profile around 71 m depth (equivalent of core 3) in detail throughout the manuscript (i.e. within the comparison of the non-crustal Pb Ortles data with the Arctic Pb ice core data (section 3.2), and within the application of the COPRA model (section 4) and the Dansgaard-Johnsen model (section 5).*

While we believe we have now clarified why a discussion of an age reversal would not be justified in the context of this manuscript, we report within the text why continuity at this depth is likely. Specifically:

In section 3.2 we now mention that in general the excellent match between Alto dell'Ortles and the Russian Arctic record at that depth/age provides strong evidence of the stratigraphic and temporal continuity of the basal ice core record.

In section 4, linked to the COPRA model, we already report the concept of maximum resolvable differences for the 14 C sequences which are stratigraphically coherent, considering their uncertainties.

*Another major concern I have is the traceability of the proceeding of depth and time scale matching using d18O and Pb profiles, i.e. for the depth scale matching of the three cores and dating of the common Ortles depth scale between 1907 CE and ~200 BCE by matching the non-crustal Pb Ortles profile with the Pb data of a well dated Arctic ice core.*

*Although the authors describe their proceeding very detailed, it is not easy to follow their proceeding and arguments.*
*To improve that:*
*1. A table or a scheme should be added in section 2, which gives an overview on: which parameters are available in which parts of which of the three cores, and their respective use in view of the dating procedure. Since different reference depth scales are used in the Figures (Fig 1. reference depth from core 2, Fig 2 reference depth of core 3, Fig 5 reference depth of core 1)*

*that will be an important guidance for the reader throughout the different sections of the manuscript.*

We thank the reviewer for this excellent suggestion. A table describing the parameters used for dating in each of the three cores is now provided within the main text.

 *In addition, a SI table would be very helpful which reports the used tie point depths of the three cores in absolute and water equivalent depths for each core.*

We now also report in the supplementary Table2 the depths in m and m we of the tie points in the original cores from where they were obtained. It will be possible to obtain any other depth by interpolation using the map depths (illustrated graphically in Fig. 1) that will also be uploaded on the public repository.

*2. Figure 1 is as it stands not useful for the reader since the matching of d18O profiles is not traceable. It would be better to use an illustration type as used in Fig. 2 means keep the individual depth scales and highlight tie point and their connections between the cores. Hereby prescind the principal tie points used to anchor the depth scale matching and their connections between the cores, vice versa the refinement tie points set afterwards. In addition, the y axis should be extended, in order that the variations of the d18O profiles become more obvious, especially in the lowest part of the cores.*

We have now fully revised Fig. 1 visualizing the stable isotopes records before and after the match, allowing the traceability of this operation. We have also extended the Y axis in panel b to make the d18O variations in the lowest part of the cores more visible.

*3. Figures 2, 4, S2, S3, S4 should be changed. As it stands, the reliability of the depth scale matching via the comparison of the non-crustal Pb profile cannot be assessed by the reader since in the above-mentioned illustrations the non-crustal Pb data were logarithmized or logarithmic axes are used to present the Pb data. In addition the Pb and the non-crustal Pb data are not made available for the reader (at least I did not find them). Therefore, to make the reasoning of the Pb profile matchings of the three cores and between Ortles and the Arctic core traceable for the reader, no logarithmic data and y axis scales should be used in Fig4, Fig S2, S3, S4.*

Because of the very large range of Pb values, using a linear scale would make visible only the highest values while it would make invisible all the features linked to the lower Pb values. The reliability of matching could not be assessed in this way. As a compromise, the log of the Pb values is used. In this way all the large and small features adopted to select the tie points remain visible for the reader and allow the best possible assessment of the matching.

We understand the point raised by the reviewer however and in our revised version we have adjusted the Y scale of the bottom panel in Fig. 4, Fig. S2, S3 and S4 to better illustrate the correlations resulting from the match.

*Among other things, this would allow:*

- *to see clearer at which point the two Ortles Pb records agree or differ between core 1 and core 3. Differences seem to be up to half a log scale and over up to one m depth which makes up several hundred years at this depth (e.g. around 65, 68 and 70 m depth of core 3). Since for the moment, only one of the profiles is chosen for the age matching with the Arctic ice core, this is an important point to be discussed.*

Different Pb concentration levels in the two cores are visible using the log scale. These are due to the different acid leaching time between continuous flow analyses (CFA) using online acidification (core #3) and the discrete analyses using conventional pre-acidification of the aliquots (core #1). Differences are larger for low Pb concentration levels probably because these are characterized by crustal Pb that, unlike anthropogenic Pb, is less acid-leachable. Different methods do not affect the Pb features used for matching the cores. This point is now discussed within the main text.

In addition, about 1 m difference in depth between the same layers is relative. While it is correct that at this depth a 1 m difference within the same core corresponds to a few 100 years, it is much less of a concern when considering the same depth interval between the two individual sites (about 10 meters apart from each other). In this later case, a difference in the average annual net accumulation rate less than 2% between the two sites does explain well the offset in depth. Resolving these differences in depth is the entire point of this paragraph.

- *to give the reader the opportunity to see the quality and reliability of the matching between the Ortles and the Arctic (and the CG) non-crustal Pb profiles. E.g. the authors state in line 1 of page 10 "In general, the match between Alto dell'Ortles and SZ is excellent, except during 1650-1900 CE ..." however Fig 4, Fig.S2, Fig. S3 and suggest that the matching is rather arbitrary especially after ~ 600 CE the course over time of the Ortles non-crustal Pb and Arctic Pb differs significantly.*

First of all we note that, overall, the linear correlation between the Ortles and the Arctic Pb records is r=0.58 which is significant at p < 0.01. This is quite compelling considering the distance and different conditions of the Mt. Ortles and the Arctic drilling sites. This correlation is now mentioned within the text.

Fig 4 and Fig S2-S3-S4 have now been revised to better show the quality and reliability of the Pb matching. In general we note that, while some degree of arbitrary choices is unavoidable in this kind of wiggle matching, in our study this is minimized thanks to the 14C dating that constrains the matching within the uncertainty of these absolute measurements. This is a kind of remarkable and unique guidance that highly limits the arbitrary choices made during this procedure.

As we already pointed out in our manuscript, it is correct that the matching in the 1650-1900 CE time period is not excellent. This is mostly due to the decoupling of the *relative* Pb levels in the two records, Ortles and Arctic (much lower in the Ortles core during that period, see Fig 4 and S2, probably due to snow erosion during the LIA, as discussed in section 5). However, when adjusting the Y axis for this relative decrease, the trends of the two records can be better reconciled and a better matching is observed (please zoom panel in Fig. 4 and Fig S3). This idea is supported by the comparison with the independent CG record that, starting with 1350 CE, shows a trend that is consistent with the Ortles core until 1800 AD (Fig. S2 and S3). In any event, the conservative large uncertainty adopted during the LIA interval (10 % of the age, 20-30 years) should account for any kind of less accurate matching during this period.

*As it stands (logarithmic scale) I am not convinced that the Pb matching with the Arctic core is reliable enough to shift the age of the 14C data to the limit of their 68% probability, and to neglect the 14C sample at 1361 ± 204 CE in section 4.*

The Pb match at around that time is now better illustrated in the revised Fig. S3. We would also like to clarify that the revised age scale passes only two 14C sample dates at its 68% probability limit (out of 10), which are the one at 1361 ± 204 CE and sample #103 which was already excluded in TC2016 (as already discussed in detail in the reply above; see also Fig. 6c). In any case, as already mentioned within the text, we do take the arising uncertainty of the dating for this section into account by providing a dating error at around 10% of the respective age to acknowledge the discrepancy between the Pb-matched and the 14C ages. This uncertainty is fully included in our final dating derived by the application of the Monte Carlo simulation (COPRA).

*It will be also very important to see, whether the assumption of a depth age disturbance would improve the Pb matching agreement of Ortles and the Arctic ice core or not.*

We hope we have now clarified that there is no evidence to assume an age disturbance. In particular, matches of the Pb records of Mt. Ortles cores #1, #3 and the Arctic core are excellent at around 71 m depth (around 350 CE; Fig 4 and S2). We thus feel we do not have sufficient evidence to justify a speculative alteration of the excellent match obtained using the Pb records at that depth. Again, in any event, the final dating comes with an uncertainty of around 10%.

*An alternative to showing the Pb and non-crustal Pb data on non-logarithmic scales and making them public within this manuscript, might be to join the refined dating addressed here to the aimed environmental discussion of the Pb data in a common manuscript.*

We carefully considered this possibility. However, we have concluded that the original dating method presented and all the details needed to properly describe the revised chronology largely justify a stand-alone paper to present the revised chronology. Another paper linked to the environmental interpretation is in preparation, making the remaining part of the data set available on the designated public repository.

*Other comments:*
*The application of the Dansgaard-Johnsen 1D model (see Dansgaard et al., 1996) on the revised depth-age relationship at the end of the manuscript (section 5) should be shortened. Only synthesized essential information of the model, the different input parameter, the outcome and the discussion of the results stay in the manuscript. Model description, input parameter determination of glaciological observations should be put in the SI.*

We believe  the ice-flow modeling section deserves an important amount of space in this manuscript. Future efforts in combining ice core dating and ice-flow modeling are much needed in order to increase our understanding of the behavior, dynamics and temporal variations in small scale alpine glaciers. Aiming at providing some outline of the major questions, challenges and perhaps the different perspectives coming from related research fields (ice core research, glaciology), we decided to include some of the basic concepts to make it understandable for an extended community. Particularly, we consider the input parameter determination to be essential in this context. The excellent agreement between measured and modeled horizontal flow velocities shown in Figure 7 is, to our best knowledge, likely unique. The fact that a simple 1D model is able to accurately reproduce the observations further provides strong support to our conclusions and we feel it needs to be kept within the main text. The different approaches used to find the model parameters (showing that they do result in very comparable values) are also a crucial part of this section. However, to accommodate the suggestion of the referee to reduce the length of the manuscript we have now

shortened the modelling section by transferring the major part of the general model description to the supplementary information.

*For the comparison of non-crustal Pb Ti is used at CG and Rb at Ortles (Figures S2, S3, S4). Why Ti was not used as well at Ortles for consistency? Please report in the text the crustal values used for the Figures. Also discuss and provide errors in calculating non-crustal Pb. How large are the changes of Rb with age in the Ortles ice (please report the Rb profile in the Supplementary material).*

Analyses of the CG and the Ortles cores were part of different projects conducted in different labs, times and by different people. As a consequence Ti was not determined in the Ortles ice cores while Rb was not determined in the CG cores. However, the selection of the crustal reference (Ti, Rb, Al, Ba etc.) does not typically influence the extent of the crustal corrections and thus we are confident that trends in non-crustal Pb levels in Ortles and CG are fully comparable. The crustal ratios used for the correction of the crustal component of Pb are i) Pb/Rb=0.51 for Ortles core #3 (obtained from a deep (69.55-69.94 m) Ortles core #3 section characterized by the lowest Pb concentrations in the record) and ii) Pb/Ti= 0.00545 (obtained from Wedepohl GCA 1995) for the CG core. These values are now reported within the text.

The error in calculating the non-crustal Pb and Ti is governed by the uncertainty in their determination by ICP-SFMS (typically the order of 5%) that provides a maximum uncertainty in the order of 10% in their respective non-crustal components.

The Rb concentration profile in core #3 is displayed below in linear and log scale.

[Figure]
* * *
*References which are not cited in the manuscript but which should:*

*Hoffmann, H., Preunkert, S., Legrand, M., Leinfelder, D., Bohleber, P., Friedrich, R., & Wagenbach, D. (2018). A new sample preparation system for micro-14C dating of glacier ice with a first application to a high alpine ice core from Colle Gnifetti (Switzerland). Radiocarbon, 60(02), 517–533. https://doi.org/10.1017/rdc.2017.99.*

*Preunkert, S., J.R. McConnell, H. Hoffmann, M. Legrand, A. Wilson, S. Eckhardt, A. Stohl, N. Chellman, M. Arienzo, & R. Friedrich (2019) Lead and antimony in basal ice from Col du Dome (French Alps) dated with radiocarbon: A record of pollution during antiquity, Geophys Res Lett, doi:10.1029/2019GL082641.*

These two papers are now cited within the manuscript.